# Effect of Ethanolic Extract of *Vernonia amygdalina* on the Proliferation, Viability and Function of Mouse Induced Pluripotent Stem Cells and Cardiomyocytes

**DOI:** 10.3390/plants12051108

**Published:** 2023-03-01

**Authors:** Arlette Nguemfo Tchankugni, Mpoame Mbida, Jürgen Hescheler, Filomain Nguemo

**Affiliations:** 1Research Unit of Biology and Applied Ecology, Department of Animal Biology, Faculty of Science, University of Dschang, Dschang P.O. Box 067, Cameroon; 2Center for Physiology and Pathophysiology, Faculty of Medicine and University Hospital Cologne, University of Cologne, 50931 Cologne, Germany

**Keywords:** natural products, medicinal plants, *Vernonia amygdalina*, pluripotent stem cells, cardiomyocytes, cytotoxicity

## Abstract

*Vernonia amygdalina* (*V. amygdalina*) leaves are commonly used in traditional medicine around the world for the treatment of a plethora disorders, including heart disease. The aim of this study was to examine and evaluate the cardiac effect of *V. amygdalina* leaf extracts using mouse induced pluripotent stem cells (miPSCs) and their cardiomyocytes’ (CMs) derivatives. We used a well-established stem cell culture to assess the effect of *V. amygdalina* extract on miPSC proliferation, EB formation and the beating activity of miPS cell-derived CMs. To study the cytotoxic effect of our extract, undifferentiating miPSCs were exposed to different concentrations of *V. amygdalina*. Cell colony formation and EB morphology were assessed using microscopy, whereas the cell viability was accessed with an impedance-based method and immunocytochemistry following treatment with different concentrations of *V. amygdalina*. Ethanolic extract of *V. amygdalina* induced toxicity in miPSCs, as revealed by a decrease in cell proliferation and colony formation, and an increase in cell death at a concentration of ≥20 mg/mL. At a concentration of 10 mg/mL, the rate of beating EBs was observed with no significant difference regarding the yield of cardiac cells. In addition, *V. amygdalina* did not affect the sarcomeric organization, but induced positive or negative effects on miPS cell-derived CMs’ differentiation in a concentration-dependent manner. Taken together, our findings demonstrate that the ethanolic extract of *V. amygdalina* affected cell proliferation, colony forming and cardiac beating capacities in a concentration-dependent manner.

## 1. Introduction

Worldwide, today’s healthcare systems face serious concerns, especially with the emergence of new infectious diseases such as COVID-19 and Ebola. Populations, especially in developing countries, are still relying on traditional treatments, which constitute the main and sometimes the only source of healthcare. These treatments are accessible, culturally acceptable, trusted and affordable. Because of skyrocketing health-care costs, the affordability of most of these traditional medicines over their conventional counterparts makes them even more attractive.

Natural products remain an important source of drugs used to sustain health, and the search for active molecules against various and incurable diseases continues to grow. Moreover, there is increasing use of herbal medicines for the treatment of a plethora of diseases, and demands for more scientific evidence for their efficacy and safety remain a huge challenge [1], as active constituents, contaminants and drug interactions can result in toxic effects, and there is limited scientific knowledge on the effect of various medicinal plants. The possibility to rapidly identify undesirable and/or desirable compounds in natural product extracts is a critical step in the drug screening and development process. Accordingly, researchers are intensively examining whether plants commonly used traditionally in medicine could be beneficial for heath and, if so, what their mechanisms of action are. In addition, they are working hard not only to elucidate their side effects, but also to determine the appropriate dosages and the biological properties, as well as to elucidate the best extraction approaches and preservation.

*V. amygdalina* is a small shrub that grows in tropical Africa and Asia, and is commonly referred to as “bitter leaf” due to its characteristic bitter taste. *V. amygdalina* can regenerate rapidly, and grows to a height of 2–5 m. Its green leaves are rich in fats, proteins, fibres, minerals, amino acids, carbohydrate and vitamins [2]. Although the green leaves are mostly consumed as food across tropical Africa, most parts of this soft-wooded shrub are widely exploited for their ethnomedicinal uses and, as reviewed by [3], it is alleged to be effective in the treatment of various diseases such as those associated with the cardiovascular system. As previously reported, extract from *V. amygdalina* includes several bioactive chemical compounds such as 6β,10β,14β trimethylheptadecan-15 α-olyl-15-O-β-D-glucopyranosyl-1,5 β olide, glucuronolactone, 11 α-hydroxyurs-5,12-dien-28-oic acid-3 α,25-olide, 10-geranilanyl-O-β-D-xyloside, 1-heneicosenol O-β-D-glucopyranoside, apigenin, luteolin (3′,4′,5,7tetrahydroxyflavone), vernolide, hydroxyvernolide, 3′-deoxyvernodalol, vernodalol, diterpene (ingenol-3-angelate), vernomygdin, 4-methylumbelliferone, cephantharin, cryptolepine, isocryptolepine, neocryptolepine, courmarins, vernolepin and vernoniosides [4]. This plant has been shown to be a bio-safe antioxidant that could be relevant in the fight against oxidative stress and free radicals, which may play a role in heart disease, cancer and other diseases [5]. Nevertheless, solid scientific evidence is still needed for the functional claims important to traditional medicine to be fully accepted by the scientific community. It is, therefore, of importance to evaluate the pharmacological activities of medicinal herbs in current use, elucidate their mechanisms of action and test for any toxic effects. In light of this, we made use of pluripotent stem cells to examine the biological properties of different concentrations of ethanolic extract of *V. amygdalina*, on the one hand, to investigate its cytotoxic effect to highlight any hidden toxic activity, and on the other, to examine its effects on the spontaneous beating activity of CMs after short- and long-term exposure. We used a real-time cell analysis (RTCA) system, which allows a real-time, non-invasive and label-free analysis of cell functions. This system uses cell index values resulting from impedance changes across the cardiac monolayer to indirectly measure cell viability and electrical activity, and has been used to screen drug action on cardiac function and cell proliferation, as well as to estimate the risk of drug safety [6,7,8].

Our experiment provided insights into the therapeutic role of *V. amygdalina* in the management of cardiovascular diseases in traditional medicine. Moreover, it also revealed efficient and effective approaches to evaluate the cytotoxicity or side effects of any existing and/or new drug candidates.

## 2. Materials and Methods

### 2.1. Plant Extract Preparation

The leaves of *V. amygdalina* were collected in the Bamboutos Division, West Region of Cameroon, and identified by the Cameroon National Herbarium (Yaoundé) using a voucher specimen registered under the Reference N° 9535/SRF. The collected plant material was dried in shade at ambient temperature for about two weeks, after which it was blended, and the powder obtained was used for the preparation of the extract as previously described [9].

One hundred (100) g of stored powder was macerated in 1.5 L of ethanol 95% for 72 h at room temperature. The mixture was stirred daily to permit better extraction of the active ingredients. The solution was sieved and filtered through a cotton layer and a filter paper of pore size 2.5 μm. The filtrate was evaporated in a rota vapor (Buchi-R-124) at 82 °C for 8 h. The extract obtained was then poured into a large Petri dish and placed in a TITANOX brand oven ventilated at 50 °C to allow complete evaporation of the solvent. Forty-eight (48) h later, the ethanol extract obtained was weighed and kept in a refrigerator at 4 °C for further processing.

Phytochemical analysis of crude and fractions of *V. amygdalina* revealed the presence of flavonoids, alkaloids, anthraquinone, steroid, phenol, phytate, oxalate, cyanogenic glycoside, tannins and saponins as described previously [10,11].

### 2.2. Mouse iPS Cells Culture and Treatment

The murine iPS cell (miPSC) line TiB7.4, as described previously [12], was used. Briefly, this miPSC line was maintained on irradiated mouse embryonic fibroblasts (MEF) in Dulbecco’s modified Eagle’s medium (Gibco) supplemented with 15% fetal bovine serum (FBS), 1 × non-essential amino acids, 2 mM l-glutamine, 50 μM 2-mercaptoethano, and 1000 U/mL leukemia inhibitory factor (LIF; ESGRO, Chemicon International). Murine embryonic fibroblasts were prepared from transgenic C57BL6 mice carrying the neomycin resistance gene at embryonic day 14.5 and inactivated through mitomycin C treatment. The cells were passaged regularly at day 2 or 3 using trypsinization (0.05% trypsin/EDTA) and 0.5  ×  10^5^ cells were added to a 6 cm dish with preplated murine embryonic fibroblasts (0.8  ×  10^5^/dish). At various concentrations, ethanolic extract of *V. amygdalina* was applied to the culture media. miPSCs were differentiated as previously described [13] in the presence of different concentrations (0, 10 and 40 mg/mL) of the extract to examine its cytotoxic effects on EBs’ morphology. The extract was added on the second day of differentiation and this was repeated every three days until the analysis. Morphological assessment and beating activity of the EBs were performed using phase contrast microscopy, and detection of the presence of beating clusters using microscopy and immunostaining were among some of the analyses performed on differentiated cardiac cells.

### 2.3. xCELLigence Real-Time Cell Analyzers Platform

The xCELLigence real-time cell analyzer (RTCA) ACEA Biosciences, San Diego, CA, USA), which uses a non-invasive high-resolution impedance readout, was used to assess in real time the functional activity (attachment, morphology, rate proliferation and size) of miPSCs growing inside the wells of the E-Plate Cardio 96 (ACEA Biosciences, San Diego, CA, USA) before and after the application of varying concentrations of *V. amygdalina* ethanol extract. Each well of the E-Plate was coated with 5 μg/mL fibronectin and incubated for 3 h at 37 °C, and freshly passaged miPSCs were seeded at a density of 25,000 cells/well with IMDM medium supplemented with 20% FBS. The plate was placed in and incubator at 37 °C with 5% CO_2_ for half an hour to allow for cell attachment. Following this initial cell attachment, the E-Plate Cardio 96 was placed in the xCELLigence RTCA Cardio reader (ACEA BIOSciences Inc., San Diego, CA, USA) and the effect of *V. amygdalina* on proliferating iPSCs was monitored for at least 10 days. CI measurements were taken every 10–15 min, with a medium change every second day beginning from day 2 when treatment first began. The number of attached cells, their morphology and beating activities were reflected in the CI measurements. A schematic description of the xCELLigence real-time workflow is show in Figure 1 below.

### 2.4. Microelectrode Array and Extracellular Field Potential Recordings

A microelectrode array (MEA) system with a Multichannel Systems 1060-Inv-BC amplifier and data acquisition system (Multichannel Systems, Reutlingen, Germany) was used for extracellular field potential (FP) recordings as described previously [14]. The MEA electrode consists of 60 titanium-nitride electrodes each with gold contacts (30 μm diameter) arranged in an 8 × 8 electrode grid and an inter-electrode separation of 200 μm. FPs can be recorded at a sampling rate of up to 50 kHz. Standard measurements were taken at a sampling rate of 2 kHz in serum-free IMDM medium. For the extracellular recordings, beating EBs were plated on MEA culture dishes pre-coated using a 1:200 solution of fibronectin (1 mg/mL) or gelatin (0.1%). All preparations were allowed to attach on MEA plates by incubating for 1–2 days at 37 °C and 5% CO_2_ before FP recordings. Temperatures were maintained at 37 °C during all recordings. Different concentrations of *V. amygdalina* (in mg/mL) were bath-applied to the MEA chambers containing the tissues. Baseline recordings were taken for 3 min followed by the application of different concentrations (from lowest to highest) of *V. amygdalina* diluted in serum-free medium. Recordings were taken for 3 min per concentration. MEA data were analyzed using the Spike 2 version 7 software (Cambridge Electronic Design, Cambridge, England) with specialized macros written in-house. These tools allow for the analysis of MEA signals for field potential frequencies, FP duration and amplitude.

### 2.5. Cell Viability, Cytotoxicity and Apoptosis

Alterations in structural integrity or metabolic pathways as a result of drug activity are important indicators of cell viability and cytotoxicity that may or may not be directly related to cell death. In addition to xCELLigence CI analysis, the trypan blue dye exclusion method and phase contrast microscopic monitoring of miPSC colony formation were used to assess viability, cytotoxicity and apoptosis in miPSCs before and after treatment with *V. amygdalina*.

It has long been established that live cells possess intact cell membranes that exclude certain dyes, such as trypan blue, eosin or propidium, whereas dead cells do not. Prior to the dye exclusion test to determine the number of viable cells present in a cell suspension and as previously described [15], miPSCs were incubated with increasing concentrations of *V. amygdalina* (0 to 80 mg/mL) for 24 h, washed with DMSO to remove the compound and then allowed to recover in fresh medium for 2 h. These miPSCs were resuspended in PBS, directly mixed with 0.4% Trypan blue solution for ten minutes and then counted in an EVE automatic cell counter (NanoEntek, VWR, Munich, Germany) to determine whether cells had taken up or excluded the dye. Experiments were performed at least 3 times and the mean of these results taken.

### 2.6. Quantification of Cell Nuclei with ImageJ

Images of control and *V. amygdalina*-treated miPSCs were converted to 8-bit images, then auto-thresholded by the “Make Binary” function using the default method. Overlaying nuclei were then separated with the “Watershed” function. Thereafter, the “Analyze Particles” function was used to analyze nuclear morphology.

### 2.7. Statistics

Statistical analyses were performed using Student’s t test for grouped data (in case of two groups) or one-way analysis of variance (in case of multiple groups). *p* values < 0.05 were considered statistically significant. Unless otherwise stated, error bars represent standard error of the mean (SEM) values. Statistics were calculated using GraphPad Prism Version 4.00 for Windows, GraphPad Software, San Diego, CA, USA.

## 3. Results

### 3.1. Effect of Ethanolic Extract of V. amygdalina on Pluripotent Stem Cells

In the first set of experiments, the effective concentration of the methanol extract of *V. amygdalina* against the cytotoxicity of undifferentiated miPSCs and miPSC-CMs was examined. Cells were exposed to different concentrations of *V. amygdalina* and monitored for up to two weeks using an xCELLigence RTCA Cardio Instrument (Figure 1A) as described in the methods section. To examine the effect of *V. amygdalina* on miPSCs’ proliferation, we first determined the cell index (CI) of cells cultured with 0, 0.6, 1.25, 2.5, 5, 10, 20, 40 and 80 mg/mL of methanol extract of *V. amygdalina* after 12 h and 24 h treatments. As shown in Figure 1B, the methanol extract of *V. amygdalina* affected the proliferation of the cell in a concentration- and time-dependent manner compared to that in the control condition. A statistically significant difference in CI was observed after 12 h of treatment with concentration ≥ 1.2 mg/mL, and was further pronounced after 24 h of treatment. The CI showed a relative increase from 2.7 ± 0.1 to 2.95 ± 0.2 (n = 3, *p* > 0.05) in control conditions, whereas at the highest *V. amygdalina* concentration, the CI was significantly decreased from 2.7 ± 0.1 to 0.09 ± 0.01 and from 2.95 ± 0.12 to 0.11 ± 0.02 after 12 h and 24 h treatment, respectively (Figure 1C, n = 3, *p* < 0.05).

### 3.2. Effect of Ethanolic Extract of V. amygdalina on miPSCs’ Viability

The impact of the methanolic extract of *V. amygdalina* was further examined on the viability of pluripotent stem cells after 24 h of incubation using a trypan blue exclusion assay (Figure 2). Cell viability is expressed as a percentage of the total cell number. A decrease in cell viability was observed after 24 h in a concentration-dependent manner with a maximum inhibitory effect observed at 80 mg/mL, the highest tested concentration. The results presented in Figure 2 revealed non-significant (*p* > 0.05) changes in the viability of cells exposed to the extract at a concentration of ≤10 mg/mL. However, the concentrations of 40 and 80 mg/mL significantly reduced miPSCs’ viability by 80.0 ± 2.4% and 63.7 ± 8.8% (*p* < 0.05), respectively, compared to the control group with 95.9 ± 1.7% viable cells, suggesting possible toxicity at high concentrations.

### 3.3. Effect of Ethanolic Extract of V. amygdalina on miPSCs’ Colony Formation and Nuclei

The cytotoxic effect of ethanolic extract of *V. amygdalina* was further examined by examining cells colonies’ morphology and measuring their area. The data revealed that the cells treated with *V. amygdalina* showed a significant increase in the number of colonies, but presented a remarkable reduction in area and size (Figure 3A,B) in a concentration-dependent manner. In control groups, large colonies of cell were observed (388.7 ± 40.8 µm). At a concentration of 1.25 mg/mL, the extract did not provoke a significant change in the diameter of the cell colonies (338.4 ± 75.6 µm, *p* > 0.05), whereas at 10 and 40 mg/mL, the diameters of the colonies were significantly (*p* < 0.05) reduced to 162.2 ± 29.1 µm and 70.0 ± 13.8 µm, respectively (Figure 3B). To gain insight into the morphological changes caused by the extract, we further examined its effect on cells’ number and nuclear area. The cells treated with *V. amygdalina* at 40 mg/mL for 48 h were compared with untreated cells (controls). Figure 4A–F shows images of cells’ nuclei from the control and *V. amygdalina* (40 mg/mL)-treated group. Our data analysis revealed a low number of cells (Figure 4G) with a large nuclear area (Figure 4H) in the *V. amygdalina* (40 mg/mL)-treated group in comparison with the control, which is a clear characteristic sign of cells having undergone apoptosis.

### 3.4. Impact of Ethanolic Extract of V. amygdalina on Cardiac Differentiation of Pluripotent Stem Cells

Mouse induced pluripotent stem cells (miPSCs) were differentiated into cardiomyocytes (CMs) by EB formation via mass culture protocol in the absence (control) and presence of different concentrations (10 and 40 mg/mL) of *V. amygdalina*. As shown in Figure 5A, the EBs generated under ethanolic extract of *V. amygdalina* exhibited the opposite effect in terms of morphology and growth properties. The low concentration of 10 mg/mL induced a significant increase in EBs’ size as compared to control EBs and those generated under high concentration (data not shown). The percentage of beating EBs generated in the presence of 10 mg/mL of *V. amygdalina* after 12 days of differentiation was significantly higher than those from control conditions (Figure 5B). However, a significant reduction in the percentage of spontaneously beating EBs was observed under 40 mg/mL *V. amygdalina* treatment after 12 days of differentiation. In addition, the beating rate (Figure 5C) was also increased under 10 mg/mL (156.0 ± 12.1 beats/min, *p* < 0.05) and relatively decreased under 40 mg/mL (145.2 ± 21.15 beats/min, *p* > 0.05) of *V. amygdalina* in comparison to the control (148.9 ± 14.3 beats/min).

To elucidate the mentioned effect of *V. amygdalina* on the differentiation of miPSC towards CMs, we examined the cardiomyogenesis efficiency and the integrity of CMs generated under 10 mg/mL and 40 mg/mL of *V. amygdalina*. The quantification of single GFP-positive cells (CMs) from day 12 EBs revealed a comparable number of CMs in the control and *V. amygdalina*-treated conditions (data not shown). As shown in Figure 5D, the higher magnification of single CMs obtained from the *V. amygdalina*-treated (40 mg/mL) conditions revealed the presence of well-defined striated sarcomeric structures representing typical cardiac cross-striated patterns as stained by sarcomeric α-actinin. This result suggests that CMs generated under *V. amygdalina* conserve at least their structure and morphology, despite changes in the percentage of beating EBs.

Moreover, the acute influence of *V. amygdalina* on spontaneously beating cardiac cluster was further investigated. Figure 6A shows a representative picture of an MEA plate containing attached miPSC-CM tissue. The results revealed that *V. amygdalina* affects the miPSC-CM clusters’ beating frequency and amplitude in a biphasic dose-dependent manner (Figure 3B–D) marked by an initial increase in FP frequency up to concentration 10 mg/mL, and a corresponding decrease in FP for higher concentrations. As shown in Figure 3C,D, the application of *V. amygdalina at* concentration ≤ 10 mg/mL led to an increase in the FP frequency of 24% (4.5 ± 0.5 vs. 3.6 ± 0.4 Hz for control) and FP amplitude of about 20% (*p* < 0.05, n = 3). Higher concentrations of *V. amygdalina* instead showed an opposite effect to low concentrations, with a dose-dependent decrease in FP frequency. At 20 mg/mL, a decrease of about 3% and 7% in FP frequency and amplitude, respectively, was observed, whereas 40 mg/mL induced a significant decrease in FP frequency (−36%) and amplitude (−21%) (*p* < 0.05, n = 3). The action of *V. amygdalina* was partially recovered after washout, confirming the effect of this extract on miPS cell-derived CM clusters.

## 4. Discussion

Phytochemicals remain an important source of compounds with chemical properties effective in treating diseases. Worldwide, different plants and herbal extracts are utilized as alternative remedies in the treatment of various diseases [16]. Moreover, the utilization of bioactive compounds from plants can be useful in the discovery of new medicines. In the face of challenging diseases for which no effective treatments exist, plants could offer insights into novel therapeutic options for control and treatment measures [16]. Nevertheless, evidence of the efficacy, mechanisms of action and toxic effects of most of these plant compounds remain an important and active area of biomedical research. There is always a need for better candidate models (cells and/or organisms) and approaches for in vitro and in vivo studies. In addition to understanding the cell specification process, pluripotent stem cell differentiation is a useful model for in vitro developmental cardiotoxicty testing and to screen new compounds [17]. Secondary metabolites such alkaloids, terpenoids, phenolics and others obtained from medicinal plants are active compounds found in nature and are the most common source of pharmaceuticals and nutraceuticals drugs currently marketed. Here, we evaluate the safety and pharmacological activities of the ethanolic extract of *V. amygdalina* by using pluripotent stem cells and their differentiated derivatives. The differentiation of pluripotent stem cells (PSCs) towards EB formation into specific lineages is connected to the size and morphology, and it recapitulates different aspects of organogenesis and a high degree of self-organization, as we recently reported [18]. The EB size and shape are important parameters that affect cell proliferation, lineage specification and commitment in vitro [16]. Cells were exposed or cultured with different concentrations of *V. amygdalina* for different durations. We found that ethanolic extract of *V. amygdalina* significantly induced toxicity of miPSCs, as revealed by a decrease in cell proliferation and colony formation, and an increase in cell death after treatment with a high concentration (≥20 mg/mL). Further, we demonstrated that *V. amygdalina* exhibits a biphasic chronotropic activity on miPS cell-derived CM clusters, with an initial low dose (10 mg/mL) inducing positive chronotropic activity (increase in FP beating activity) followed by a corresponding significant decrease in chronotropic activity at doses above 20 mg/mL. However, *V. amygdalina* revealed no effect on the sarcomeric structures of miPS cell-derived CMs, as indicated by immunostaining. The phytochemical screening of VA revealed the presence of flavonoids, saponins, alkaloids, tannins, phenolics, terpenes, steroidal glycosides, triterpenoids and different types of sesquiterpene [19,20], which can be postulated for several pharmacological activities including antioxidant, cardiotonic, antimicrobial, antihypertensive, anticancer and analgesic characteristics. Numerous studies have investigated the pharmacological activity of *V. amygdalina*, which has been shown to exhibit potent antibacterial, anticancer and anti-diabetic activities and is stated to be cytotoxic and cardioprotective [21,22]. Our results are also in agreement with previous studies demonstrating the inhibition of cell proliferation, as evidenced by the reduced colony formation and increase in cell death following V. amygdalina exposure. Thus, the ethanolic extract *V. amygdalina possesses* an anticancer potential. Our data also show that, depending on the concentration, the ethanolic extract of *V. amygdalina* induces positive and negative chronotropic effects on cardiomyocytes. In fact, muscarinic receptor activation plays a crucial role in the parasympathetic regulation of heart function due to the expression of the muscarinic receptors throughout the cardiovascular system; muscarinic agonists are capable of pharmacologically producing both inhibitory and stimulatory effects on the heart [14], despite the fact that they are not usually implicated in regulating the parasympathetic nervous system [23]. In vivo, muscarinic receptors can also modulate the electrical and mechanical activity of all cardiac subtypes, as well as the conduction of electrical impulses through the heart. The action of the ethanolic extract of *V. amygdalina* observed on cardiac cells may be, at least in part, implicated in both their decrease and increase in beating activity, where its vasodilatory effect mediate a decrease in heart rate, as suggested previously on aortic rings isolated from Sprague Dawley rats [24]. Endothelium-dependent relaxing factors (EDRFs), which include prostacyclin (PGI_2_) and nitric oxide (NO), followed by K^+^ channel openers) and subsequently M3- and ß2-receptor signaling pathways were the main mechanism for generating this vasorelaxant effect. Therefore, our results are in line with this observation obtained using miPS cell-derived CMs.

Moreover, some peptides from *V. amygdalina* were shown to be effective inhibitors of mitogen-activated protein kinase (MAPK) [5], a cytosolic signaling proteins that becomes activated after specific phosphorylation [25], which has a significant impact on cardiac gene expression, contractility, extracellular matrix remodeling and inflammatory response in the heart [26]. These biological events are the conclusion of signaling cascades mainly by four MAPK subfamilies made up of extracellular signal-regulated kinases (ERK1/2), c-Jun NH2-terminal kinases, p38 kinase and big MAPK (BMK or ERK5) [27], which may act individually or in combination. A previous study revealed that the ERK/MAPK signaling pathway promotes cell proliferation and hinders apoptosis [28]. In addition, during the cardiac differentiation process of P19, three distinct MAP kinases, namely JNK, p38 and ERK1/2, are coordinately activated, and have been shown to contribute to proliferation, cell specification and differentiation [29], as well as the regulation of cardiac function.

In conclusion, the current study showed that the ethanolic extract from the leaves of *V. amygdalina* increases and decreases the beating activity of miPS cell-derived CMs in a concentration-dependent manner. To decrease and enhance the spontaneously beating frequency, the extract may work through the muscarinic and ß-adrenergic pathways, respectively, as suggested previously [14]. In addition, the ethanolic extract of *V. amygdalina* can induce cell cytotoxicity in a dose-dependent manner. It seems to be extremely cytotoxic at concentrations > 20 mg/mL, inhibiting cell proliferation and the beating activity of CMs. Chemicals found in *V. amygdalina* modulate cardiac chronotropic activity either singly or in combination, and may, therefore, be useful in creating antiarrhythmic medications. However, it can also be potentially harmful if not used properly. Thus, despite its broad in vitro and in vivo pharmacological activity using different cells and organ models from various origins, additional studies and human clinical trials are required to determine the effective and safe dosages for the aforementioned disorders.

## Figures and Tables

**Figure 1 plants-12-01108-f001:**
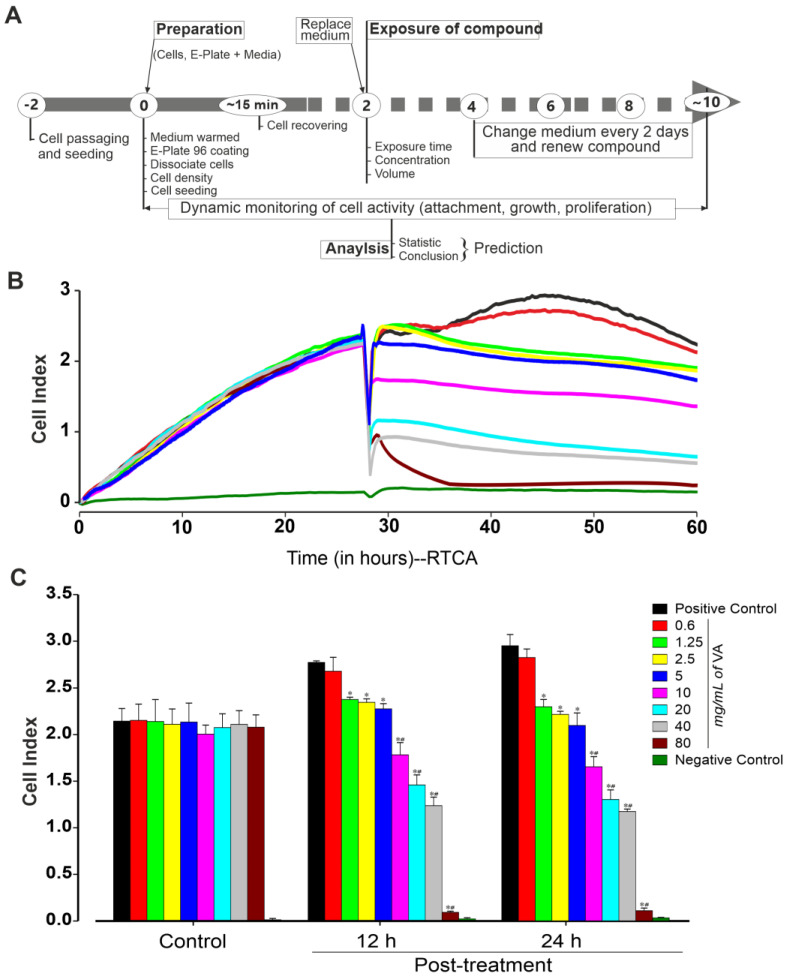
Experimental workflow for preparing, culturing, growing and recording compounds’ effect using the xCELLigence RTCA Cardio system. (**A**) On day 0, confluent miPSCs were dissociated and plated (at a density of 25,000 cells/well) into an E-Plate 96-well previously coated with fibronectin. Different concentrations of *Vernonia amygdalina* were generally added 24 h post-plating. Every two days, medium and/or compound were replaced with a fresh one. Cell activity (attachment, morphology, rate proliferation and size) was monitored for at least 10 days. (**B**) Exemplary recordings showing changes in cell index in the presence of different concentrations of VA as compared to positive (cell without treatment) and negative (well without cells) control conditions. (**C**) Peak cell index values before (24 h post-plating) and after treatment with VA at the concentrations of 0 (positive control), 0.6, 1.25, 2.5, 5, 10, 20, 40 and 80 mg/mL for 12 and 24 h. Data are presented as the mean ± SEM of three independent experiments. * *p* < 0.05 and ^#^
*p* < 0.05, significantly different compared with control and previous low concentration, respectively.

**Figure 2 plants-12-01108-f002:**
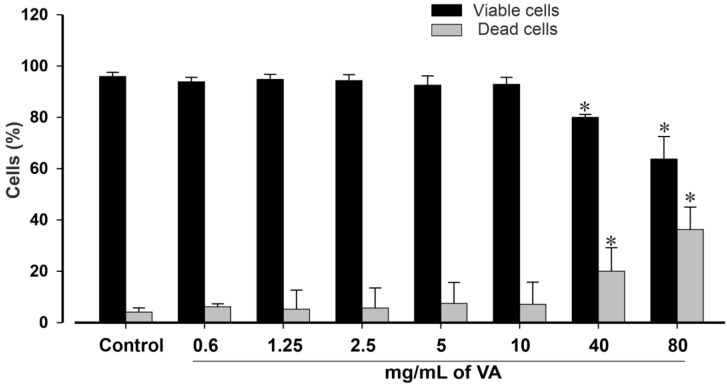
Statistical analysis of viable and dead cells in controls and VA-treated conditions. *Vernonia amygdalina* reduces the viability of mouse iPSCs in a concentration-dependent manner. Cells were incubated with increasing concentration of VA (0 to 80 mg/mL) for 24 h. Thereafter, the samples were washed to remove the compound. The cells were then allowed to recover in fresh medium for 2 h. After wise, Trypan blue exclusion method was used to assess cell viability. Suspension of each sample was mixed with 0.4% Trypan blue solution for 10 min and cells were counted in an EVE automatic cell counter (NanoEntek, VWR, Munich, Germany). Experiments were performed at least 3 times and results are presented as the mean ± SEM, and are statistically significant for * *p* < 0.05.

**Figure 3 plants-12-01108-f003:**
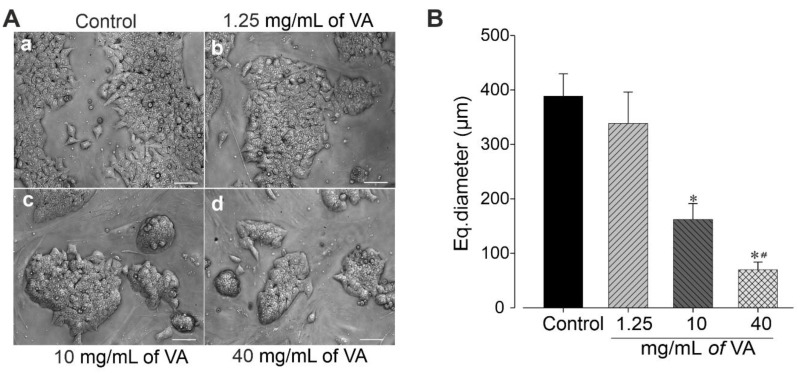
Effect of VA treatment on miPSC colonies’ formation 2 days after seeding. (**A**) Phase contrast images of cell colonies obtained under control (**a**) and in the presence of various concentrations of VA at day 2 in culture; 1.25 mg/mL (**b**), 10 mg/mL (**c**) and 40 mg/mL (**d**). (Scale bar: 200 μm). These images are a representative result of three independent experiments. (**B**) Dimensional properties of miPSC colonies quantified by equivalent diameter. Data are presented as the mean ± SEM of three independent experiments. * *p* < 0.05, significantly different compared with control treatment and ^#^ with previous concentration Scale bars: 100 µm.

**Figure 4 plants-12-01108-f004:**
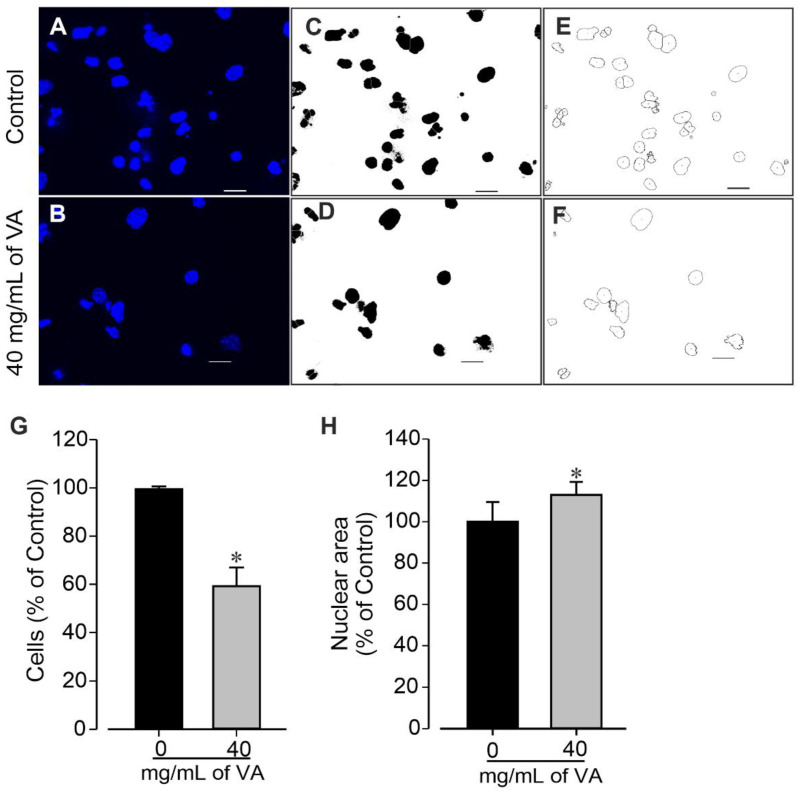
Quantification of cell nuclei using ImageJ as described previously. Cell cultures in the absence (control, 0 mg/mL of VA) and presence of 40 mg/mL of VA (treated) were observed using fluorescence microscopy after nuclei staining with Hoechst 33258. (**A**,**B**) Original image of nuclei from control (**A**) and VA-treated group of cells (**B**). (**C**,**D**) Nuclei thresholded image of control and treated cells. (**E**,**F**) Nuclei watershed boundaries of cells from control (**E**) and 40 mg/mL l VA-treated group (**F**). Images are representative of three independent samples. (**G**,**H**) Number of cells (**G**) and nuclear area (**H**) of control (0 mg/mL) and 40 mg/mL VA-treated cells. Data are presented as the mean ± SEM of three independent experiments. * *p* < 0.05, significantly different compared with control (0 mg/mL VA) condition. Scale bars: 50 µm.

**Figure 5 plants-12-01108-f005:**
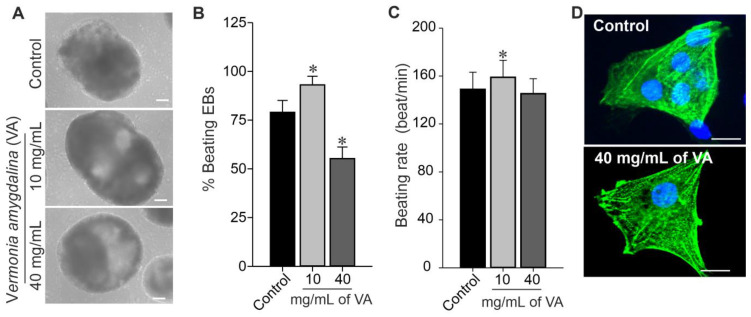
Effect of VA on CMs. Cardiac differentiation of miPSCs, undifferentiated miPSCs differentiated into CMs by inducing formation of EBs containing spontaneously beating cardiac cells. (**A**) Representative transmitted-light microscopy images of individual EBs at day 12 of differentiation cultured under different concentrations of VA. (**B**,**C**) Quantitative analyses of the percentage of beating EBs and EB beating rate differentiated under control and different concentrations of VA. (**D**) Images of cardiomyocytes derived from untreated (control, above) and miPSCs treated (below) with 40 mg/mL *Vernonia amygdalina* for 24 h. Cells were labeled with antibodies to detect sarcomeric actin (green), and nuclei were stained with Hoechst 33,342 (blue). Data are presented as the mean ± SEM of three independent experiments. * *p* < 0.01, significantly different compared with control (0 mg/mL VA) condition. Scale bars: 100 µm.

**Figure 6 plants-12-01108-f006:**
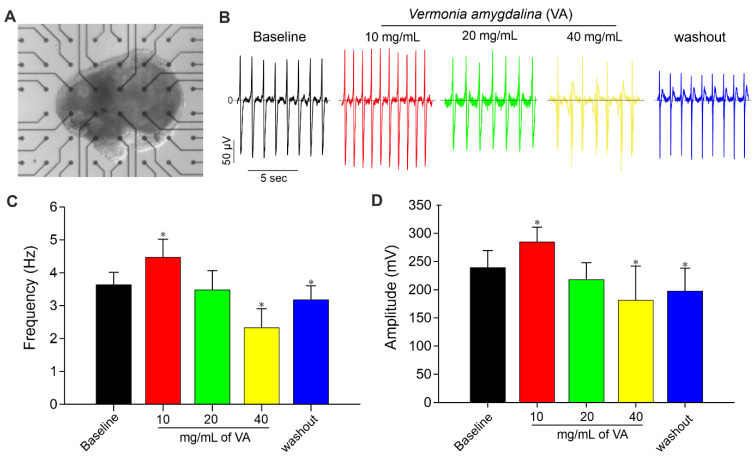
MEA Recording of cardiac clusters and effect of *V. amygdalina*. (**A**) Viable and spontaneously beating iPSC-derived cardiac cluster attached to MEA chamber during field potential (FP) recordings. (**B**) Representative FP traces of one electrode from MEA analysis of spontaneously beating iPSC-derived cardiac cluster demonstrating the effects of different concentrations (10, 20 and 40 mg/mL) of *V. amygdalina*. (**C**,**D**) Statistical analysis of FP frequencies (**C**) and amplitude (**D**) of baseline and *V. amygdalina* in MEA measurements of cardiac cluster. Data are shown as mean ± SEM of at least three independent experiments. * *p* < 0.05, significant difference between baseline and *V. amygdalina*-treated cardiac cluster.

## Data Availability

Not applicable.

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
