# Peer review of "Effect of Ethanolic Extract of Vernonia amygdalina on the Proliferation, Viability and Function of Mouse Induced Pluripotent Stem Cells and Cardiomyocytes"

_plants, 2023, doi:10.3390/plants12051108_

Round 1

Reviewer 1 Report

The Authors evaluated pharmacological effects of ethanolic extract obtained from Vernonia amygdalina on pluripotent stem cells differentiated into cardiomyocytes. I recommend publication of the manuscript after minor corrections:

1)      The Authors should describe the limitations of their study (e.g., resulted from the use of mouse cells)

2)      The Authours should better describe possible application of the obtained results (instead of only laconic statements concerning the use of the extract as a support in the treatment of cardiological diseases)

Author Response

We appreciate your thoughtfulness in allowing us to revise our work. We are grateful to the reviewers for their insightful, thorough, and helpful comments and ideas. We take all their concerns seriously. By implementing the reviewers' suggestions as much as we could into the final version, we have rectified and improved the manuscript.

We address each reviewer's specific concerns in the sections that follow, and we think that the paper has significantly improved as a result. The changes tracked version of the manuscript highlights every modification in red.

Reviewer 1

The Authors evaluated pharmacological effects of ethanolic extract obtained from Vernonia amygdalina on pluripotent stem cells differentiated into cardiomyocytes. I recommend publication of the manuscript after minor corrections:

1)      The Authors should describe the limitations of their study (e.g., resulted from the use of mouse cells)

Response:

We thank the reviewer for his comments

2)      The Authors should better describe possible application of the obtained results (instead of only laconic statements concerning the use of the extract as a support in the treatment of cardiological diseases).

Response:

We thank the review for this concern; we revised the manuscript accordingly by incorporating the possible application of our result. But additional experiment and clinical trials will be absolutely necessary before any recommendation.

Add into the discussion:

Thus, despite it´s broad in vitro and in vivo pharmacological activity using different cells and organs models from various origins, additional studies and human clinical trials are required to determine effective and safe dosages of the aforementioned disorders

Reviewer 2 Report

The authors use an extract without any characterization of its chemical composition. This makes repetition of this work very complicated.  There are many faciltiies that run pay per sample LC/MS or better LC/HRMS analyses on complex extracts. Samples of the screened compounds should be reported in the manuscript or Supporting/Supplemental information.  Ideally, one can also include an NMR spectrum on the extract.

As reported it is virtually impossible to properly repeat the data presented here as one as absolutely no ability to replicate the extract presented.

For this reviewer this is an ABSOLUTE requirement as most facilities only charge the cost of <$100 to do this work.

Google search: contract LC/MS or better LC/HRMS (HRMS=high resolution mass spectrometry) services

Minor edits and suggestions mostly in the Figures and figure quality have been provided.

This was a very well written manuscript and well done.

Author Response

We appreciate your thoughtfulness in allowing us to revise our work. We are grateful to the reviewers for their insightful, thorough, and helpful comments and ideas. We take all their concerns seriously. By implementing the reviewers' suggestions as much as we could into the final version, we have rectified and improved the manuscript.

We address each reviewer's specific concerns in the sections that follow, and we think that the paper has significantly improved as a result. The changes tracked version of the manuscript highlights every modification in red.

Reviewer 2

The authors use an extract without any characterization of its chemical composition. This makes repetition of this work very complicated.  There are many faciltiies that run pay per sample LC/MS or better LC/HRMS analyses on complex extracts. Samples of the screened compounds should be reported in the manuscript or Supporting/Supplemental information.  Ideally, one can also include an NMR spectrum on the extract.

As reported it is virtually impossible to properly repeat the data presented here as one as absolutely no ability to replicate the extract presented.

For this reviewer this is an ABSOLUTE requirement as most facilities only charge the cost of <$100 to do this work.

Google search: contract LC/MS or better LC/HRMS (HRMS=high resolution mass spectrometry) services

Response:

We appreciate the reviewer's feedback. Our study's purpose was not to define the chemical constituents of the extract, which has previously been extensively done, but to evaluate the aqueous extract (as used traditionally) on cell proliferation and cardiac cell function. We don't believe that reanalyzing the chemical composition of our extract would provide any new information. A review document describing the chemical composition of the extract is available for reviewers (see Eziuche A.Ugbogu et al, Clinical Complementary Medicine and Pharmacology, 2021). In the introduction, we We also highlighted the chemical composition of this plant in the updated manuscript.

Minor edits and suggestions mostly in the Figures and figure quality have been provided.

Response:

We thank the reviewer for the remarks. All figures have be updated according to the journal requirement.

This was a very well written manuscript and well done.

Response:

Thank you for your remark
